# Sir4 Deficiency Reverses Cell Senescence by Sub-Telomere Recombination

**DOI:** 10.3390/cells10040778

**Published:** 2021-04-01

**Authors:** Jun Liu, Xiaojing Hong, Lihui Wang, Chao-Ya Liang, Jun-Ping Liu

**Affiliations:** 1Institute of Ageing Research, School of Medicine, Hangzhou Normal University, Hangzhou 311121, China; xiaojing.hong@hznu.edu.cn (X.H.); wanglihui100@163.com (L.W.); liangchaoya1992@163.com (C.-Y.L.); 2Department of Immunology, Faculty of Medicine, Monash University, Prahran, VIC 3181, Australia; 3Hudson Institute of Medical Research, Clayton, VIC 3168, Australia

**Keywords:** cell senescence, senescence regulation, sub-telomeres, telomere binding protein, Sir4, Rif1, yeast

## Abstract

Telomere shortening results in cellular senescence and the regulatory mechanisms remain unclear. Here, we report that the sub-telomere regions facilitate telomere lengthening by homologous recombination, thereby attenuating senescence in yeast *Saccharomyces cerevisiae*. The telomere protein complex Sir3/4 represses, whereas Rif1 promotes, the sub-telomere Y′ element recombination. Genetic disruption of *SIR4* increases Y′ element abundance and rescues telomere-shortening-induced senescence in a Rad51-dependent manner, indicating a sub-telomere regulatory switch in regulating organismal senescence by DNA recombination. Inhibition of the sub-telomere recombination requires Sir4 binding to perinuclear protein Mps3 for telomere perinuclear localization and transcriptional repression of the telomeric repeat-containing RNA *TERRA*. Furthermore, Sir4 repression of Y′ element recombination is negatively regulated by Rif1 that mediates senescence-evasion induced by Sir4 deficiency. Thus, our results demonstrate a dual opposing control mechanism of sub-telomeric Y′ element recombination by Sir3/4 and Rif1 in the regulation of telomere shortening and cell senescence.

## 1. Introduction

Telomeres at the termini of eukaryotic chromosomes regulate genome stability and cell fate. Shortening of telomeres constitutes a mechanism against the proliferative potential of eukaryotes and induces DNA damage response and genome instability [1,2,3,4,5]. Recruitment of telomerase by specific proteins on telomeres rescues telomere shortening [6,7]. In the budding yeast *Saccharomyces cerevisiae*, telomerase contains the catalytic reverse transcriptase subunit Est2, intrinsic RNA template *TLC1*, two accessory protein subunits Est1 and Est3 [8,9,10,11,12], and Pop1/6/7 subunits [13]. For recruitment, two telomerase components are recognized specifically, with Est1 bound by the single-stranded telomeric DNA (ssDNA) binding protein Cdc13 in the cell cycle S phase and *TLC1* bound by the double-stranded telomeric DNA (dsDNA) binding protein Ku complex (Yku70/80) in the G1 phase [6,7,14,15]. Disruption of Yku80-*TLC1* interaction by the *yku80–135i* allele results in telomere shortening [14], but not as short as that of *yku80*Δ cells, suggesting that Yku has other functions at telomere ends, e.g., restricting the invasive action of *Exo*I exonuclease to telomere C-strand resection [16,17].

Deprotection of telomeres with telomerase deficiency leads to replicative senescence [18]. Disruption of Cdc13 interaction with Est1 [19], *EST1* deletion [20], *yku80*Δ *mre11*Δ cells [21], or *yku70*∆ *mre11*∆ cells [22] causes cell senescence. Similar to mammalian alternative lengthening of telomeres (ALT), however, yeast cells escape senescence to form two types of survivors: type I survivor characteristic of highly amplified sub-telomeric Y′ elements and type II survivor with heterogeneous telomere TG_1–3_ sequence recombination [20]. Both types of survivors require Rad52 which generally mediates homologous recombination [20,23] including the break-induced replication-mediated telomere elongation [24,25]. In addition, type I survivors require Rad51, Rad54, Rad55, and Rad57 to amplify sub-telomeric Y′ elements, and make use of the Rad59-mediated Rad52 strand annealing activity to delay the onset of senescence [26].

Telomere lengthening that takes place in the absence of telomerase via DNA homologous recombination has been investigated [20,27,28,29]. However, which telomere proteins regulate the sub-telomere recombination remains unclear. Sir4, Sir3, Rif1, and Rif2 are recruited to duplex telomeres by interacting with the carboxyl termini of Rap1 [8,9], with Sir4 further recruiting Sir2 and Sir3 to telomeres [30]. Whereas Sir4-Sir2-Sir3 ternary complex participates in chromatin silencing [30] by interacting with not only Rap1 but also Yku80 [31,32], Rif1 and Rif2 inhibit the recruitment of telomerase to telomeres [33,34]. Moreover, in G1 phase, while silencing duplex telomeres, Sir4 binding to Yku80 recruits Yku80-*TLC1* telomerase complex to the 3′ G-rich single strand telomeres [6,7], as Yku70/80 bindings to telomerase RNA *TLC1* or telomeric DNA are mutually exclusive [35].

Deletion of *YKU80* and *MRE11* results in a senescence-like phenotype even in the presence of telomerase [22], whose mechanism is not fully understood. The present study aimed to determine how telomere-proteins Sir2, Sir3, Sir4, Rif1, and Rif2 regulate senescence and sub-telomere recombination in the presence of telomerase. We observed that Sir4 deficiency rescued yku80Δ mre11Δ cell senescence by significantly increasing sub-telomere Y′ element amplification and telomere lengthening. Consistently, loss of Sir4-Mps3 interaction-mediated telomere perinuclear localization and increase in *TERRA* transcription contribute to Y′ element amplification induced by Sir4 deficiency in yku80Δ mre11Δ cells. We further showed that Rif1 deficiency, or Rif1/2 deficiencies, suppressed yku80Δ mre11Δ cell senescence by lengthening telomeres and inhibiting Y′ element amplification. Thus, our studies demonstrate that in the presence of active telomerase, Sir4 deficiency significantly increases sub-telomere Y′ element amplification to lengthen telomeres, which is amendable by Rif1 or Rif1/2 deficiencies in diverting cellular senescence in yeast.

## 2. Material and Methods

### 2.1. Strains, Oligonucleotides, and Genetic Manipulations

*Saccharomyces cerevisiae* and plasmids used in this study are shown in Appendix A. Gene knock-out (KO) mutants were constructed as described previously [36,37]. Heterozygous diploid strains with one copy gene targeted were subjected to sporulation and tetrad dissection. Gene KO cassettes were specifically amplified by PCR using high-fidelity DNA polymerase in the following plasmids: pFA6a-HIS3MX6 (HIS3MX6), pFA6a-KanMX6 (KanMX6), pYM-N15 (N15), or pFA6a-8Glyc-13Myc-HphMX6 (HphMX6). Listed in Appendix A are the PCR combinations of paired primers and corresponding plasmid-templates for amplification of gene KO cassettes. Other primers used in this study are listed in Appendix A.

### 2.2. Sporulation and Micromanipulation

Diploid strains were subjected to sporulation using a protocol described previously in SGD (http://www-sequence.stanford.edu/group/yeast_deletion_project/spo.html, accessed on 31 March 2021) with modifications. For details, refer to Appendix A.

### 2.3. Telomere Length and Y′ Element Analyses by Southern Blot

Yeast strains were grown on YPD or selective synthetic medium plates as indicated in a 30 °C incubator for 3–4 days. Single colonies were successively streaked on fresh YPD or selective synthetic plates every 3–4 days, cells from the plates were collected, washed, and stored at −80 °C freezer. Genomic DNA isolation and telomere length measurement were performed by Southern blotting as described previously [38].

### 2.4. Cell Senescence Analysis

For cell senescence analysis in yeast liquid cultures, spores with indicated genotypes were inoculated into 500 µL fresh YPD liquid medium and incubated overnight to prepare saturated cultures at 30 °C 220× rpm. A calculated quantity of cells for each genotype was diluted into 10 mL fresh YPD medium to give an initial concentration of A600_nm_ = 0.01. Cell density (A600_nm_) was measured by a spectrophotometer, calculated and re-diluted into 10 mL fresh YPD medium approximate every 20 h. Cells of the liquid culture were collected and washed with water for DNA extraction. For cell senescence analysis on YPD plate, different colonies of spores for each genotype were streaked every 48 h and followed by photographing and re-streaking on fresh plates.

### 2.5. Quantitative Analysis of Genomic Y′ Element Copy Numbers

Y′ element relative copy numbers in genomic DNA was analyzed by real-time quantitative PCR using primers specific to amplify consensus DNA fragment in all the 19 Y′ elements of 17 telomeres of 13 *S.cerevisiae* chromosomes [39]. Data are results from 3–5 biological replicates and expressed as averages with S. E. M. indicated by error bars.

### 2.6. Statistical Significance Calculation

Statistical calculation was done by using the two-sided student’s t test. The *p* value considered statistically significant was set to be smaller than 0.05.

## 3. Results

### 3.1. Sir3/4, But Not Sir2, Deficiency Rescues Senescence

To investigate if Sir2/3/4 play a potential role in the senescence of *yku80*∆ *mre11*∆ cells, we constructed diploid strains heterozygous for *yku80*∆, *mre11*∆, *sir2*∆, *sir3*∆, and *sir4*∆. The strains were sporulated and tetrad dissected, respectively (Appendix A), *yku80*∆ *mre11*∆ cells grew normally at 1st streakout, significantly slow displaying senescence phenotype at the 3rd streakout (Figure 1A), consistent with the senescence phenotype observed in *yku70*∆ *mre11*∆ cells [22]. However, deletion of *SIR4* in *yku80*∆ *mre11*∆ cells showed comparable growth rates to that of *mre11*∆ and *mre11*∆ *sir4*∆ cells (Figure 1A), so did *SIR3* deletion, but not the *SIR2* deletion, in *yku80*∆ *mre11*∆ cells (Appendix A). In liquid senescence assay, *yku80*∆ *mre11*∆ cells exhibited significantly reduced proliferative rate and subsequently succumbed to senescence at the 3rd passage as compared to *mre11*∆ *sir4*∆, *yku80*∆ *sir4*∆, and *mre11*∆ cells (Figure 1B). Sir4 deficiency in *yku80*∆ *mre11*∆ cells prevented senescence leading to cell density comparable to that of *mre11*∆ and *mre11*∆ *sir4*∆ cells (Figure 1B). Consistently, the cell density of *yku80*∆ *mre11*∆ *sir4*∆ cells was significantly increased compared with that of yku*80*∆ *mre11*∆ cells (*p* = 6.78 × 10^−9^) (Figure 1B). *yku80*∆ *mre11*∆ and *yku80*∆ *mre11*∆ *sir2*∆ cells exhibited slow growth rate to the extent by and large indifferent from that of *yku80*∆ *mre11*∆ senescent cells following six successive passages (*p* = 0.59) (Appendix A). Sir3 deficiency caused senescence evasion in *yku80*∆ *mre11*∆ cells similar to that of Sir4 deficiency (*p* = 8.86 × 10^−13^) (Appendix A). Together, these results demonstrated that the senescent phenotype induced by the *yku80*∆ *mre11*∆ deficiencies is circumvented by the deprivation of Sir3 or Sir4, but not Sir2.

Consistently, subculturing WT, *yku80*∆ *mre11*∆, *yku80*∆ *mre11*∆ *sir2*∆, *yku80*∆ *mre11*∆ *sir3*∆, *yku80*∆ *mre11*∆ *sir4*∆ cells by proportional dilution and plating onto fresh YPD plates to observe the growth phenotype of single colonies showed that *yku80*∆ *mre11*∆ and *yku80*∆ *mre11*∆ *sir2*∆ cells were less with a mixture of small and large colonies compared to that of WT cells (Appendix A), but deletion of *SIR3* or *SIR4* inhibited the senescence of *yku80*∆ *mre11*∆ cells (Appendix A). To determine if Sir3 or Sir4 deficiency rescues *yku80*∆ *mre11*∆ cell senescence through the same pathway, we constructed diploid strain that was heterozygous for *yku80*∆ *mre11*∆ *sir3*∆ *sir4*∆ (Appendix A). The results showed that the cell density of *yku80*∆ *mre11*∆ *sir3*∆ *sir4*∆ cells are indistinguishable to that of *yku80*∆ *mre11*∆ *sir3*∆ and *yku80*∆ *mre11*∆ *sir4*∆ cells following six successive passages (Appendix A). These data show that Sir3 or Sir4 deficiency rescues *yku80*∆ *mre11*∆ cell senescence by the same pathway.

### 3.2. Rad51 Is Required for Senescence Evasion Induced by Sir4 Deficiency

We next determined whether Rad52 is required for rescuing senescence by Sir4 deficiency. Diploid strain heterozygous for *yku80*∆, *mre11*∆, *sir4*∆, and *rad52*∆ was sporulated and tetrad dissected (Appendix A). But *yku80*∆ *mre11*∆ *rad52*∆ and *yku80*∆ *mre11*∆ *sir4*∆ *rad52*∆ cells senesced and barely grew compared with that of *yku80*∆ *mre11*∆ cells (Appendix A). Liquid senescence assay using these spores obtained consistent results, showing extremely slow proliferation for the *yku80*∆ *mre11*∆ *rad52*∆ and *yku80*∆ *mre11*∆*sir4*∆ *rad52*∆ cells in liquid culture following six successive passages (Appendix A). These results demonstrated that Sir4 deficiency-rescued senescence requires Rad52 in *yku80*∆ *mre11*∆ cells.

To determine if Rad51-mediated sub-telomere Y′ element recombination is required for Sir4 deficiency-induced rescue of senescence, we constructed the diploid strain heterozygous for *yku80*∆, *mre11*∆, *sir4*∆, and *rad51*∆ (Appendix A). Significantly, *yku80*∆ *mre11*∆ *rad51*∆ and *yku80*∆ *mre11*∆ *sir4*∆ *rad51*∆ cells grew extremely slow with cell senescent phenotype at the first streakout comparable to that of *yku80*∆ *mre11*∆ cells at the third streakout, suggesting that Sir4 deficiency-rescued senescence requires Rad51 (Figure 1C). In the liquid senescence assay, *yku80*∆ *mre11*∆ *rad51*∆ and *yku80*∆ *mre11*∆ *sir4*∆ *rad51*∆ cells barely proliferated characteristic of cell senescence at the first passage when compared with that of *yku80*∆ *mre11*∆ and *yku80*∆ *mre11*∆ *sir4*∆ cells, and *yku80*∆ *mre11*∆ *rad51*∆ cells reached a cell density that was comparable to that of *yku80*∆ *mre11*∆ cells (Figure 1D). Consistently, *yku80*∆ *mre11*∆ *sir4*∆ *rad51*∆ cells did not reach the cell density of *yku80*∆ *mre11*∆ *sir4*∆ cells at the sixth passage, further confirming that Rad51-mediated sub-telomere Y′ element recombination is indispensable to the telomere maintenance of *yku80*∆ *mre11*∆ *sir4*∆ cells (Figure 1D). These results demonstrated that Sir4 deficiency-rescued cell senescence requires Rad51 in *yku80*∆ *mre11*∆ cells.

Rad59 facilitates the acquisition of Y′ elements with short telomeres [26]. To determine if Rad59 is required for Sir4 deficiency to rescue senescence, we constructed the diploid heterozygous for *yku80*∆, *mre11*∆, *sir4*∆, and *rad59*∆ (Appendix A). *yku80*∆ *mre11*∆ *rad59*∆ cells grew extremely slower than that of *yku80*∆ *mre11*∆ cells on the plate, similarly *yku80*∆ *mre11*∆ *sir4*∆ *rad59*∆ cells also grew slower than *yku80*∆ *mre11*∆ *sir4*∆ cells following six successive streakouts (Appendix A). In the liquid medium, *yku80*∆ *mre11*∆ *rad59*∆ cells proliferated more slowly at the first three successive passages but reached comparable cell density at the sixth passage when compared with *yku80*∆ *mre11*∆ cells (Appendix A). In addition, *yku80*∆ *mre11*∆ *sir4*∆ *rad59*∆ cells proliferated to an extent similarly to that of *yku80*∆ *mre11*∆ cells at the first three successive passages, and reached a comparable cell density to that of *yku80*∆ *mre11*∆ *sir4*∆ at the sixth passage (Appendix A). These data suggested that Sir4 deficiency-induced senescence-evasion is partially dependent upon Rad59 in *yku80*∆ *mre11*∆ cells.

### 3.3. Rif1, Rif1/2, But Not Rif2 Deficiency Suppresses Senescence

To investigate a potential role of Rif1 and Rif2 inhibited telomerase lengthening in senescence, we constructed diploid strains heterozygous for *yku80*∆, *mre11*∆, *rif1*∆, *rif2*∆, and *sir4*∆ with or without the *yku80–135i* allele (Appendix A). As expected, deficiency of Rif1 or both Rif1/Rif2 lengthened telomeres by telomerase recruitment, rescuing senescence of *yku80*∆ *mre11*∆ cells following six successive streakouts, whereas *yku80*∆ *mre11*∆ *rif2*∆ cells grew as slowly as that of *yku80*∆ *mre11*∆ senescent cells along with six successive streakouts (Figure 2A and Appendix A). However, the cell densities of *yku80*∆ *mre11*∆ *sir4*∆ and *yku80*∆ *mre11*∆ *rif1*∆ were comparable, and the cell density of *yku80*∆ *mre11*∆ *sir4*∆ *rif1*∆ cells was in-between those of *yku80*∆ *mre11*∆ cells and *yku80*∆ *mre11*∆ *sir4*∆ cells in liquid senescence assay (*p* = 0.26) (Figure 2B), suggesting that Rif1 deficiency compromises Sir4 deficiency-rescuing senescence in the *yku80*∆ *mre11*∆ cells by lengthening telomeres. In addition, the cell density of *yku80*∆ *mre11*∆ *rif2*∆ cells was significantly lower than that of *yku80*∆ *mre11*∆ cells following six successive passages (*p* = 0.007) (Figure 2C) and the cell density of *yku80*∆ *mre11*∆ *sir4*∆ *rif2*∆ cells was significantly lower than that of *yku80*∆ *mre11*∆ *sir4*∆ cells (*p* = 0.005), suggesting that Rif2 deficiency alone is not sufficient to rescue the senescence of *yku80*∆ *mre11*∆ cells. Moreover, the cell densities of *yku80*∆ *mre11*∆ *sir4*∆ *rif1*∆ *rif2*∆ penta mutant and *yku80*∆ *mre11*∆ *sir4*∆ triple mutant cells were comparable following six successive passages (*p* = 0.6) (Figure 2D), and the cell density of *yku80*∆ *mre11*∆ *rif1*∆ *rif2*∆ cells was significantly higher than that of *yku80*∆ *mre11*∆ cells (*p*= 3.94 × 10^−5^) (Figure 2D), consistent with the data that deficiency of Rif1, or Rif1 plus Rif2, compromises the effect of Sir4 deficiency on rescuing senescence.

### 3.4. Rif1 and Rif1/2 Deficiencies Lengthen Telomeres of yku80∆ mre11∆ and yku80∆ mre11∆ sir4∆ Cells

To investigate the mechanism by which Sir4 and Rif1 deficiencies rescue the senescence of *yku80*∆ *mre11*∆ cells, and Rif1 deficiency sabotages the effect of Sir4 deficiency on rescue *yku80*∆ *mre11*∆ cell senescence, we examined telomere length and Y′ element intensity by Southern blot of spores with or without Rif1/2, Sir3/4, Yku80, and Mre11. Notably, Rif1 and Rif2 deficiencies produced longer telomeres after six successive streakouts (Figure 3A, lanes 3–4 versus lane 2), and the longer telomeres were eliminated by *SIR4* and *MRE11* deletions (Figure 3A, lanes 5–6 versus lane 3–4), consistent with the notion that Sir4-Yku80 interaction and MRX-ssDNA mediate telomere telomerase recruitment that is inhibited by Rif1 and Rif2. Deletions of *RIF1* and *RIF2* (*rif1*∆ *rif2*∆) plus the *yku80-135i* allele and *mre11*∆ resulted in a severe blockade of telomere lengthening with observable shortest telomeres (Figure 3A, lanes 7–8), consistent with the essential roles of Yku80-Tlc1 interaction and MRX-ssDNA for telomerase recruitment. However, deletion of *SIR4* led to higher amplification of the sub-telomeric Y′ element with longer telomeres than that in the *mre11*∆ *yku80*∆ cells (Figure 3A, lanes 19–20 versus 21–22; Appendix A). The sub-telomeric Y′ element amplification induced by *SIR4* deletion in *yku80*∆ *mre11*∆ cells was significantly reduced by deletions of *RIF1* and *RIF2* (Figure 3A, lanes 19–20 versus lanes 11–12, 21–22 versus 13–14), suggesting that Sir4 and Rif1/2 are reciprocally involved in an opposing negative and positive regulation of sub-telomeric Y′ element recombination respectively. Moreover, the effects of *RIF1* or *RIF2* deletion on telomere lengths and Y′ element recombination were examined after six successive streakouts (Figure 3B). *RIF1* and *RIF2* double deletions as well as *RIF1* (but not *RIF2*) single deletion suppressed Y′ element recombination in *yku80*∆ *mre11*∆ cells (Figure 3A, lanes 13–14 versus lanes 21–22; Figure 3B, lanes 29–30 versus lanes 13–14).

### 3.5. Sir4 Deficiency Enhances, whereas Rif1 or Rif1/2 Deficiency Suppresses, Y′ Element Amplification

Next, we systematically and quantitatively examined sub-telomere Y′ element amplification by qSACH [39] in the various strains with or without *SIR4* or *RIF1* deletion. Y′ element copy number significantly increased in *yku80*∆ *mre11*∆ compared with WT (*p* = 3.02 × 10^−9^) following six successive passages (Figure 4A), consistent with published results [22]. Accordingly, *yku80*∆ *mre11*∆ *sir4*∆ cells showed significantly increased Y′ element copy numbers than yku80∆ mre11∆ cells (*p* = 4.60 × 10^−4^) (Figure 4A). Inactivation of Sir3 (*p* = 4.41 × 10^−4^), but not inactivation of Sir2, recapitulated the effects of Sir4 deficiency on increasing Y′ element copy number in *yku80*∆ *mre11*∆ cells (Figure 4A). Furthermore, we found that inactivation of Rif1 (*p* = 1.19 × 10^−11^), Rif2 (*p* = 0.02), or both Rif1 and Rif2 (*p* = 6.09 × 10^−9^) significantly decreased Y′ element copy numbers compared with *yku80*∆ *mre11*∆ (Figure 4A). More interestingly, *yku80*∆ *mre11*∆ *sir4*∆ *rif1*∆ (*p* = 9.03 × 10^−6^) and *yku80*∆ *mre11*∆ *sir4*∆ *rif1*∆ *rif2*∆ (*p* = 9.63 × 10^−4^) (but not *yku80*∆ *mre11*∆ *sir4*∆ *rif2*∆ (*p* = 0.49)) cells consistently showed significantly decreased Y′ element copy numbers than *yku80*∆ *mre11*∆ *sir4*∆ cells (Figure 4A).

### 3.6. Rif1 or Rif1/2 Deficiency Compromises sir4Δ-induced Y′ Element Recombination

*yku80*∆ *mre11*∆ *rif1*∆ *rif2*∆ cells had longer telomeres than *yku80*∆ *mre11*∆ cells, suggesting that *rif1*∆ *rif2*∆ increased telomerase recruitment in *yku80*∆ *mre11*∆ (known as telomerase recruitment failure) cells. However, *yku80*∆ *mre11*∆ *sir4*∆ *rif1*∆ *rif2*∆ cells had lower Y′ element recombination that *yku80*∆ *mre11*∆ *sir4*∆ cells (Figure 4A), consistent with a role of *rif1*∆ *rif2*∆ in lengthening telomeres to suppress sub-telomere recombination. Nevertheless, Y′ element intensity in *yku80*∆ *mre11*∆ *sir4*∆ *rif1*∆ *rif2*∆ cells was still higher than that of *yku80*∆ *mre11*∆ cells, suggesting that *rif1*∆ *rif2*∆ mediated telomere lengthening incompletely suppresses the sir4∆-induced Y′ element amplification, i.e., Sir4-deficit had stronger effects on Y′ element recombination than Rif1/2-deficiencies.

### 3.7. Rad51 Deficiency Eliminates Y′ Element Amplification in yku80∆ mre11∆ and yku80∆ mre11∆ sir4∆ Cells

To further confirm that Sir4 deficiency mediated rescue of *yku80*∆ *mre11*∆ cell senescence was due to Rad52/Rad51 mediated Y′ element amplification, we examined Y′ element amplification in *yku80*∆ *mre11*∆ cells with or without *RAD52* or *RAD51* deletion. Our results showed that the Y′ element amplification in *yku80*∆ *mre11*∆ *rad52*∆, *yku80*∆ *mre11*∆ *rad51*∆, *yku80*∆ *mre11*∆ *sir4*∆ *rad52*∆, and *yku80*∆ *mre11*∆ *sir4*∆ *rad51*∆ cells was completely sabotaged (Figure 4B). These data indicate that Sir4 deficiency rescues *yku80*∆ *mre11*∆ cell senescence by increasing Rad52/Rad51-mediated Y′ element amplification.

### 3.8. Increased TERRA and Loss of Telomere Perinuclear Localization Contributes to the Y′ Element Amplification Induced by Sir4 Deficiency

PAF1 complex epigenetically inhibits telomere transcription of *TERRA* whereas paf1Δ and *ctr9*Δ cells produce higher *TERRA* in a Sir4-independent manner [40], we determined if the epigenetic regulation of telomere transcription by Sir4 might be related to sub-telomere Y′ amplification regulated by Sir4 using diploid strains heterozygous for *yku80*Δ *mre11*Δ *sir4*Δ *paf1*Δ and *yku80*Δ *mre11*Δ *sir4*Δ *ctr9*Δ (Appendix A). Spore cells carrying paf1Δ and ctr9Δ alleles had extremely slow growth phenotype (Appendix A), therefore these spore cells were streaked onto fresh YPD plates and allow them to grow 4 days. Remarkably, sub-telomere Y′ element abundance in *yku80*Δ *mre11*Δ *sir4*Δ *paf1*Δ (*p* = 1.64 × 10^−9^) cells or *yku80*Δ *mre11*Δ *sir4*Δ *ctr9*Δ cells (*p* = 1.63 × 10^−8^) was increased to the levels 3–5 folds higher than that of yku80Δ mre11Δ sir4Δ cells during six successive streakouts (Figure 5A), suggesting that *paf1*Δ/*ctr9*Δ alleles mediate the Y′ element amplification. The striking potentiating effect of Paf1/Ctr9 loss on Sir4 deficiency-induced sub-telomere Y′ element amplification reflected an intimate role of PAF1/Ctr9 epigenetic regulation of sub-telomere Y′ homologous recombination by a mechanism parallel with Sir4 in *yku80*Δ *mre11*Δ cells, highlighting a Sir4- and PAF1/Ctr9-related epigenetic program in the regulation of sub-telomere homologous recombination and telomere transcription in a coordinated manner.

To determine if Sir4-mediated telomere perinuclear localization [41,42] might be involved in the repressive mechanisms of Y′ element amplification, we constructed the diploid strains heterozygous for *yku80*Δ *mre11*Δ *sir4*Δ mps3Δ75-150 for comparisons with that of *yku80*Δ, *mre11*Δ *sir4*Δ *siz2*Δ (Appendix A). Loss of Siz2, a PIAS-like SUMO E3 ligase that sumoylates both Yku80 and Sir4 and promotes telomere anchoring to the nuclear envelope [42], did not significantly increase Y′ element amplification (*p* = 0.21) in *yku80*Δ *mre11*Δ cells following six successive passages (Figure 5B), suggesting that loss of perinuclear envelope anchoring alone is not sufficient to increase Y′ element amplification in *yku80*Δ *mre11*Δ cells. However, introduction of the mps3Δ75-150 allele in yku80Δ mre11Δ cells to specifically disrupt the interaction between Mps3 (a Sad1-UNC-84 domain protein that is required to anchor telomeres at the nuclear envelope) and Sir4 [41] significantly increased Y′ element copy numbers (*p* = 0.047) (Figure 5B), suggesting that telomere perinuclear location mediated by Sir4 interaction with mps3′s acidic domain (amino acids 75–150) plays an important inhibitory role in Y′ element amplification in the absence of Yku80 and Mre11. Furthermore, the Y′ element copy numbers in yku80Δ mre11Δ sir4Δ mps3Δ75–150 cells were significantly lower than that of yku80Δ mre11Δ sir4Δ (*p* < 0.005) but in-between of that in yku80Δ mre11Δ sir4Δ and yku80Δ mre11Δ mps3Δ75–150 cells, consistent with a notion that Sir4 deficiency and mps3Δ75–150 allele stimulate Y′ element amplification in the same pathway by releasing telomere perinuclear location through unleashing an epigenetic heterochromatin program in the deficiencies of both Yku80 and Mre11.

## 4. Discussion

We demonstrate a pivotal role of Sir4 in suppressing homologous recombination–dependent sub-telomere Y′ element amplification in a telomerase-positive sub-telomere recombination cell senescence model. Sir4 deficiency activates a significant increase in sub-telomere Y′ amplification intensity and frequency to lengthen telomeres and divert cell senescence in the yeast strains deficient of Yku80 and MRX that mediate telomerase recruitment and DNA repair pathways respectively [6,7,14,15,43]. Given the intermediate role in mediating telomerase recruitment as reported recently [6,7], Sir4 emerges to maintain telomere homeostasis by two distinct mechanisms: telomerase recruitment and suppression of sub-telomere Y′-element amplification. Although Sir2 and Sir3 are recruited to telomeres similarly by Sir4, Sir3, but not Sir2, they operate along with Sir4 to repress sub-telomere Y′ element homologous recombination, with Sir3 deficiency resembling that of Sir4 in rescuing cell senescence by promoting Y′ element amplification. Consistently, similar scenarios for the roles of telomerase recruitment of Sir2, Sir3, and Sir4 were reported previously, with Sir2 playing a minor one [6]. The function of Sir4 and Sir3 in the negative regulation of sub-telomere homologous recombination is consistent with the roles of the Sir protein complex in epigenetic silencing and maintaining telomere heterochromatin as shown previously [30]. Our findings suggest the meaningfulness of exploiting sub-telomere heterochromatin homologous recombination silencer Sir4 to link recruiting telomerase. So, when Sir4 deficiency occurs with very short telomeres, sub-telomere homologous recombination would become activated simultaneously to prevent telomere shortening for survival and proliferative potential.

The mechanism by which Sir4 deficiency triggers sub-telomere Y′ element homologous recombination appears to be mediated by unchecking the heterochromatin inhibitory control over an access of homologous recombination machineries (Figure 6A,B). Evidence supporting this conclusion is three-fold. First, we find that the paf1∆ or ctr9∆ allele that increases telomeric DNA transcription for *TERRA* significantly stimulates and potentiates Sir4 deficiency-induced sub-telomere Y′-element amplification following successive streakouts (Figure 5A), indicating PAF1 complex functions in parallel to Sir4 in silencing sub-telomere Y′ element amplification. Second, consistent with unchecking the heterochromatin network suppressing sub-telomere Y′ element homologous recombination, we find that disrupting Sir4 interaction with Mps3 to inhibit Sir4-dependent telomere perinuclear positioning activates sub-telomere Y′ element amplification (Figure 5B). These findings are consistent with previous studies that Sir4 mediates telomere anchoring at the nuclear envelope [41] and that Sir4 loss disrupts the telomere fold-back structures [44,45] redistributing telomeres with SUN domain-containing protein Mps3 interacting with Yku70/80-Tlc1-Est1. In addition, we find that Rad52, Rad51, and partially Rad59 are required in mediating Sir4 removal-induced sub-telomeric Y′-element amplification, highlighting that the mechanism of Sir4 deficiency-induced sub-telomere Y′-element amplification is not an isolated event, but belongs to an evolved overall cellular program mediated by Rad52, Rad51, and partially Rad59 in a coordinated manner with telomere transcription and repositioning from anchoring nuclear envelope. Although PAF1 suppression of telomeric DNA transcription for *TERRA* is independent of Sir4 [40], our findings that PAF1 deficiency potentiates Sir4 deficiency-induced sub-telomere Y′ amplification suggest that PAF1 and Sir4 are related in epigenetic suppressing sub-telomere homologous recombination. Given that the effects between removing Sir4 and disrupting Sir4 interaction with Mps3 converges on sub-telomere Y′ amplification, it is likely that Sir4 regulates sub-telomere Y′-element homologous recombination suppression by mechanisms involving the modulation of telomere nuclear periphery positioning at least in part in trans.

More interestingly, Rif1 and Rif2 are recruited to duplex telomeres by Rap1 that recruits Sir4, offering a potential *cis* mechanism for Rif1 and Rif2 to regulate Sir4-induced sub-telomere Y′-element homologous recombination silencing [34]. By genetic deletions of *RIF1* and *RIF2*, we demonstrate that Sir4 and Rif1 undergo functional interaction with opposing roles in negative and positive regulations of sub-telomere Y′ element homologous recombination respectively (Figure 6A,B). We show that Rif1 and Rif2 underpin sub-telomeric homologous recombination (Figure 4A) and deficiencies inhibit Sir4 deficiency-induced sub-telomeric Y′-element amplification by lengthening telomeres (Figure 3A, Figure 4 and Figure 6B). Finally, the identifications of Rif1/Rif2 in mediating, and Sir4 in suppressing, sub-telomere Y′ element homologous recombination are remarkable in that Rif1/Rif2 and Sir4/Sir3 respectively inhibits and stimulates telomerase recruitment to lengthen telomeres. The dual roles for each of the two proteins in both regulating telomerase recruitment and sub-telomere Y′ amplification suggest that Sir4 and Rif1 are used to protecting one primary mechanism of telomerase lengthening of telomeres while inhibiting another second mechanism as a backup until unless telomerase recruitment or availability is compromised.

## Figures and Tables

**Figure 1 cells-10-00778-f001:**
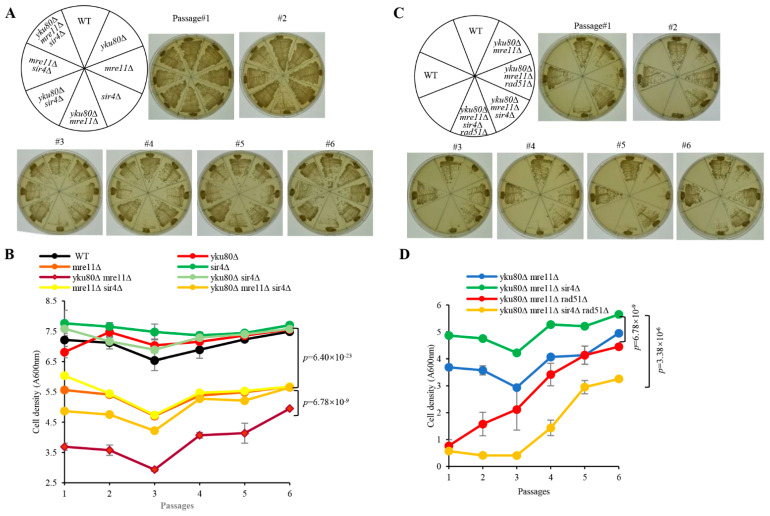
Effects of single and double deletions of *SIR4* and *RAD51* on senescence. (**A**) Spore WT, *yku80*∆, *sir4*∆, *mre11*∆, *yku80*∆ *sir4*∆, *yku80*∆ *mre11*∆, *mre11*∆ *sir4*∆, and *yku80*∆ *mre11*∆ *sir4*∆ cells were successively streaked on YPD plates every 2 days for six times. Images of photographed colony growth are from one of typical 5 experiments. (**B**) Spore cells indicated in (**A**) were cultured in liquid YPD medium at 30 °C 220× rpm with initial concentration of A600_nm_ = 0.01 for around 20 h and followed by A600 _nm_ measurement, and re-dilution was made every 20 h. Results are mean ± SEM (*n* = 5). (**C**) Spore WT, *yku80*∆ *mre11*∆, *yku80*∆ *mre11*∆ *sir4*∆, *yku80*∆ *mre11*∆ *rad51*∆, *yku80*∆ *mre11*∆ *sir4*∆ *rad51*∆ cells were successively streaked on YPD plates every 2 days for six times. Images of photographed colony growth are from one of typical 5 experiments. (**D**) Spore cells indicated in (**C**) were cultured in liquid YPD medium at 30 °C 220× rpm with initial concentration of A600_nm_ = 0.01 for around 20 h and followed by A600 _nm_ measurement, and re-dilution was made every 20 h. Results are mean ± SEM (*n* = 5).

**Figure 2 cells-10-00778-f002:**
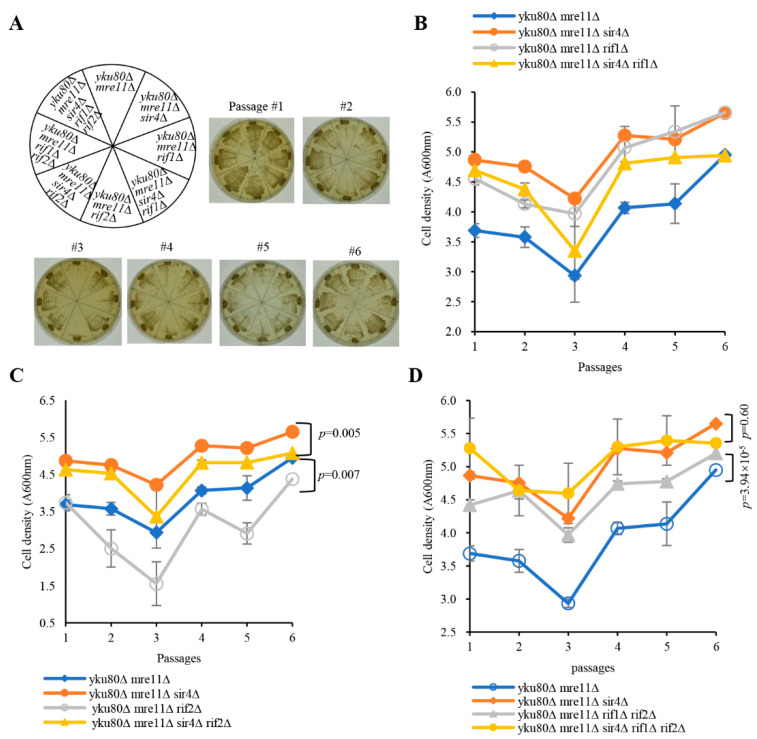
Effects of single and double deletions of *RIF1* and *RIF2* on the senescence of *yku80*∆ *mre11*∆ and *yku80*∆ *mre11*∆ *sir4*∆ cells. (**A**) Spore *yku80*∆ *mre11*∆, *yku80*∆ *mre11*∆ *sir4*∆, *yku80*∆ *mre11*∆ *rif1*∆, *yku80*∆ *mre11*∆ *sir4*∆ *rif1*∆, *yku80*∆ *mre11*∆ *rif2*∆, *yku80*∆ *mre11*∆ *sir4*∆ *rif2*∆, *yku80*∆ *mre11*∆ *rif1*∆ *rif2*∆, *yku80*∆ *mre11*∆ *sir4*∆ *rif1*∆ *rif2*∆ were successively streaked on YPD plates every 2 days for six times. Images of photographed colony growth are from one of 4–5 typical experiments. (**B**) Spores *yku80*∆ *mre11*∆, *yku80*∆ *mre11*∆ *sir4*∆, *yku80*∆ *mre11*∆ *rif1*∆, and *yku80*∆ *mre11*∆ *sir4*∆ *rif1*∆ cells. (**C**) *yku80*∆ *mre11*∆, *yku80*∆ *mre11*∆ *sir4*∆, *yku80*∆ *mre11*∆ *rif2*∆, and *yku80*∆ *mre11*∆ *sir4*∆ *rif2*∆. (**D**) *yku80*∆ *mre11*∆, *yku80*∆ *mre11*∆ *sir4*∆, *yku80*∆ *mre11*∆ *rif1*∆ *rif2*∆, and *yku80*∆ *mre11*∆ *sir4*∆ *rif1*∆ *rif2*∆ cells were cultured in liquid YPD medium with initial concentration of A600_nm_ = 0.01 for around 20 h and followed by A600 _nm_ measurement, and re-dilution was made every 20 h. Results are mean ± SEM (*n* = 4–5).

**Figure 3 cells-10-00778-f003:**
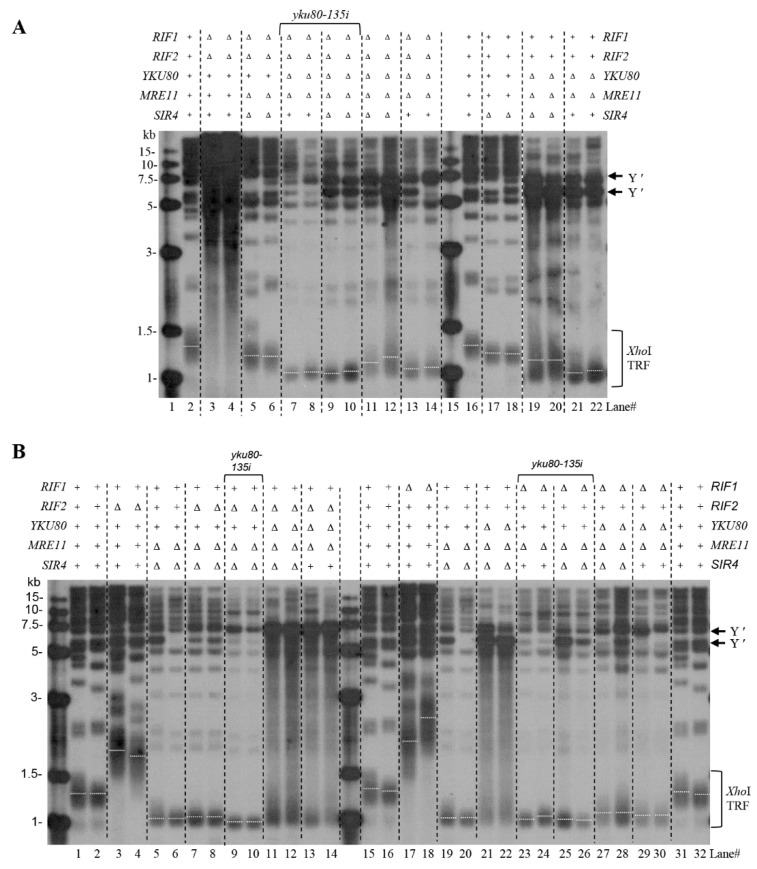
Telomere length analysis in various mutants with or without *SIR4, RIF1,* or *RIF2* deletion in *yku80*∆ *mre11*∆ cells. (**A**) Spore WT, *rif1*∆ *rif2*∆, *rif1*∆ *rif2*∆ *mre11*∆ *sir4*∆, *rif1*∆ *rif2*∆ *mre11*∆ *yku80*∆ (*yku80-135i*), *rif1*∆ *rif2*∆ *mre11*∆ *yku80*∆ *sir4*∆ (*yku80-135i*), *rif1*∆ *rif2*∆ *mre11*∆ *yku80*∆ *sir4*∆, *rif1*∆ *rif2*∆ *mre11*∆ *yku80*∆, *sir4*∆, *mre11*∆ *yku80*∆ *sir4*∆, and *mre11*∆ *yku80*∆. (**B**) WT, *rif1*∆, *rif2*∆, *rif1*∆ *mre11*∆ *sir4*∆, *rif2*∆ *mre11*∆ *sir4*∆, *rif1*∆ *mre11*∆ *yku80*∆ (*yku80–135i*), *rif2*∆ *mre11*∆ *yku80*∆ (*yku80–135i*), *rif1*∆ *mre11*∆ *yku80*∆ *sir4*∆ (*yku80–135i*), *rif2*∆ *mre11*∆ *yku80*∆ *sir4*∆ (*yku80–135i*), *rif1*∆ *mre11*∆ *yku80*∆ *sir4*∆, *rif2*∆ *mre11*∆ *yku80*∆ *sir4*∆, *rif1*∆ *mre11*∆ *yku80*∆, *rif2*∆ *mre11*∆ *yku80*∆, *sir4*∆, *mre11*∆ *yku80*∆ *sir4*∆, and *mre11*∆ *yku80*∆ spore cells after successive six streakouts on plates every 3–4 days were collected for telomere length Southern blot analysis.

**Figure 4 cells-10-00778-f004:**
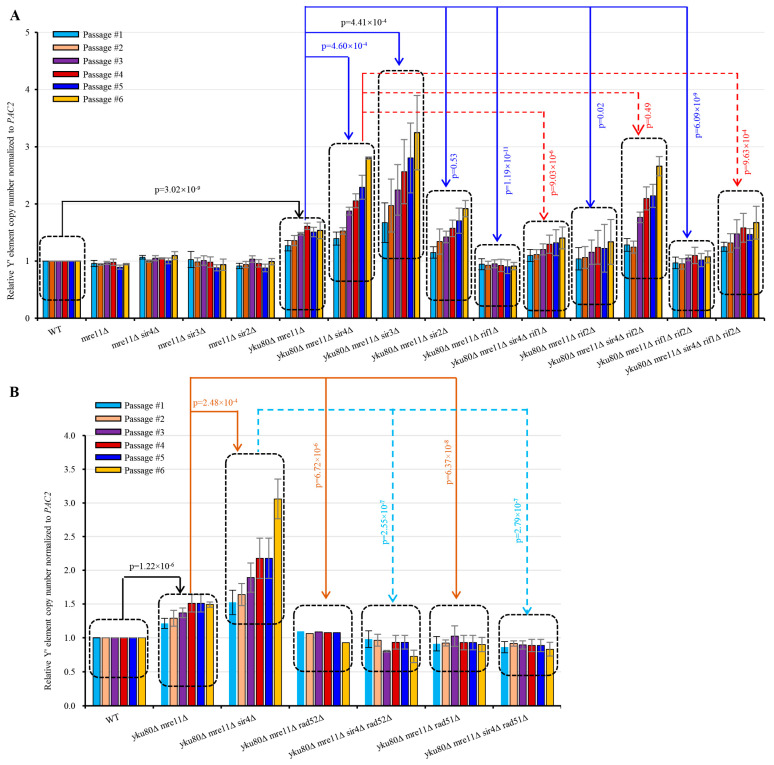
Effects of *SIR2*, *SIR3*, *SIR4*, *RIF1*, *RIF2*, *RIF1/2*, *RAD52*, *RAD51*, and *RAD59* deletions on the Y′ element amplification of *yku80*∆ *mre11*∆ and of *yku80*∆ *mre11*∆ *sir4*∆ cells. (**A**) Spore cells *yku80*Δ *mre11*Δ and *yku80*Δ *mre11*Δ *sir4*Δ with or without *RIF1*, *RIF2*, or *RIF1/2* deletions. (**B**) Spore cells *yku80*Δ *mre11*Δ and *yku80*Δ *mre11*Δ *sir4*Δ with or without *RAD51*, *RAD52*, or *RAD59* deletion were passaged successively in liquid medium every 20 h. Genomic DNAs were isolated which was followed by qSACH analyses of Y′ element copy number. Data are ± SEM (*n* = 5).

**Figure 5 cells-10-00778-f005:**
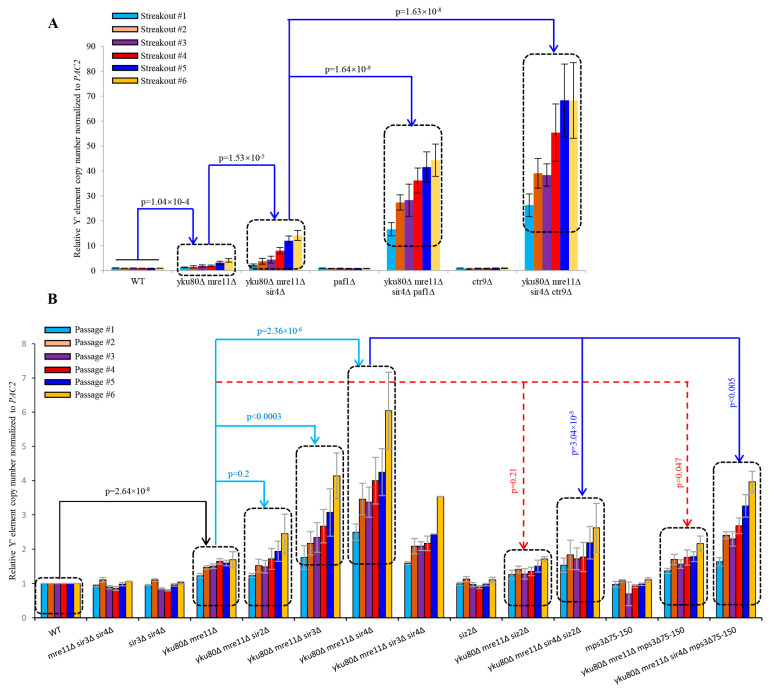
Effects of paf1∆, ctr9∆, siz2∆ or mps3∆75-150 allele on the Y′ element amplification of *yku80*∆ *mre11*∆ and *yku80*∆ *mre11*∆ *sir4*∆ cells. (**A**) Spore cells WT, *yku80*∆ *mre11*∆, *yku80*∆ *mre11*∆ *sir4*∆, paf1∆, *yku80*∆ *mre11*∆ *sir4*∆ paf1∆, ctr9∆ and *yku80*∆ *mre11*∆ *sir4*∆ ctr9∆ were successively streaked six times on YPD plates every 4 days (**B**) Spore cells with or without siz2∆ or mps3∆75–150 allele were successively passaged six times in liquid YPD medium every 20 h. Genomic DNAs were isolated and followed by qSACH analyses of Y′ element copy numbers. Data are ± SEM (*n* = 3–5).

**Figure 6 cells-10-00778-f006:**
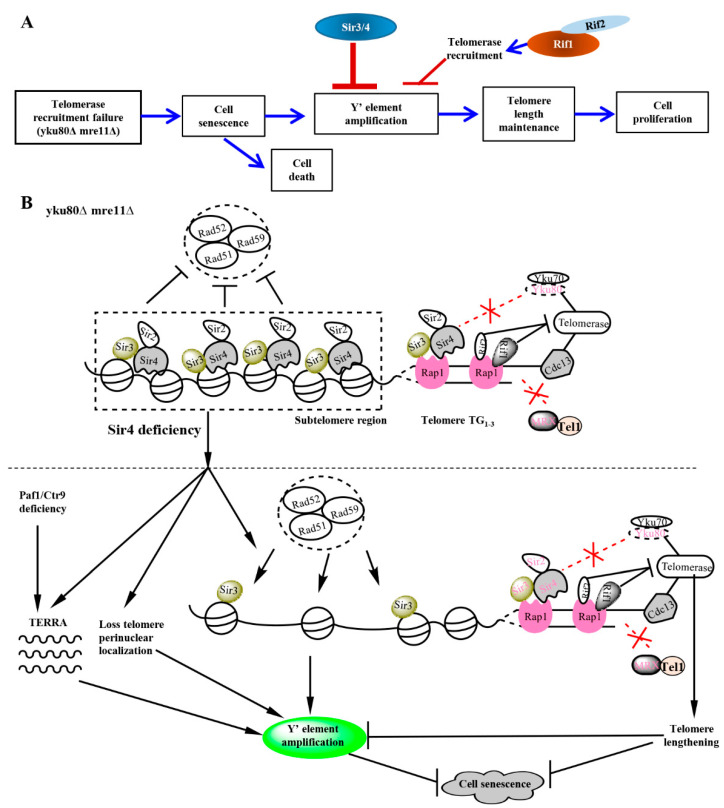
Schematic showing the roles of Sir3/4 and Rif1/2 and their relationships to the up- and down-stream elements in the rescuing pathway of senescence induced by Yku80 and Mre11 deficits. (**A**) Summary of effects of Sir3/4 and Rif1/2 on sub-telomere Y′ element amplification and senescence of *yku80*∆ *mre11*∆ cells, a telomerase-positive sub-telomere-recombination model. (**B**) A schematic picture showing the molecular roles of Rap1, Sir2/3/4, Yku80/Yku70, Rif1/2, Yku80, Mre11, Tel1, Rad52/Rad51/Rad59, telomerase, Cdc13 at sub-telomere and telomere ends. *yku80*∆ *mre11*∆ cells define a telomerase-positive sub-telomere-recombination model, following cell divisions, their telomeres do not reach critical short length but very short, some cells go through senescence characterized with reduced cell numbers during subculturing and sub-telomere Y′ element amplification, which might be due to disrupted Sir4-Yku80-Tlc1 telomerase-recruitment pathway and Rif1/2-inhibited-MRX-mediated ssDNA generation for Cdc13 loading, and pervasive action of exonuclease *Exo*I at telomere ends. Sir4 deficiency increases Y′ element amplification frequency and rescues the senescence of *yku80*∆ *mre11*∆ cells by disrupting telomere perinuclear localization, loss of telomere heterochromatin structure, and increasing *TERRA* transcription. In parallel, Paf1 or Ctr9 deficiency increases *TERRA* transcription in a Sir4-independent manner, but interestingly, further increases Y′ element amplification in sir4Δ cells. Sir4 deficiency results in changes of sub-telomere structures that might have increased the access of Rad51/Rad52/Rad59 complex mediated machinery into sub-telomere region to promote Y′ element amplification.

## Data Availability

Not applicable.

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
