# Peer review of "Sir4 Deficiency Reverses Cell Senescence by Sub-Telomere Recombination"

_cells, 2021, doi:10.3390/cells10040778_

Round 1

Reviewer 1 Report

Comments on the Ms:  cells-1153609 (Liu J.-P.)

The maintenance of telomeres ensures chromosome stability and as such is essential for continued cellular division. Given that this mechanism has importance for cancer cells in humans, it is of interest to investigate it in molecular detail. In this manuscript, telomere maintenance is probed in budding yeast cells using a variety of genetic deficiencies and measuring cellular senescence as well as subtelomeric DNA amplification. The project starts with the observations made many years ago that even in fully telomerase positive cells, a combined loss of the repair proteins yKu and Mre11 causes a phenotype that is indistinguishable from that of a loss of telomerase. It is assumed, but to date not directly shown, that this combination of mutations leads to a complete lack of recruitment of telomerase to telomeres, hence the phenotype, even though the enzyme itself still is present in cells.

Here then, the influence of additional mutations on this mechanism is assessed. The results show increased subtelomeric recombination in cells that also harbour a deletion of Sir4 or Sir3, but not with a Sir2 deletion. Furthermore, while deleting Rif1 also renders telomeric recombination more efficient, a loss of Rif2 causes the opposite. All of these effects on recombination are dependent on the Rad51 and Rad52 genes, as one would expect. Furthermore, deleting Paf1 or Ctr9 dramatically increased the subtelomeric recombination. The authors also assessed the involvement of Siz2 or Mps3 in these effects, but only detected minor differences. Together, the data are discussed as showing a critical role for Sir3/Sir4 in maintaining a normal repressive subtelomeric environment. Quantitatively, the effect of absence of PAF-complex members causes an even more important disturbance/upregulation of recombination, but it is difficult to attribute this to a specific mechanism.

Overall, the data generated are of good quality, appear robust and are well documented. Wherever possible, quantitative approaches were used, allowing for more telling conclusions. Below I outline some important suggestions for improvements at the conceptual level. However, please note that the obvious need to describe extensive and complicated genetic setups combined with a somewhat limited English makes for a difficult read. Here, I do not list all the shortcomings, it would take up too much space, but the manuscript definitely needs a major and thorough correction from a native English speaker.

Specifics:

  • I do appreciate the liquid senescence assays; the best way to measure this parameter for yeasts. However, the assay does have some pitfalls as well. Here, certain of the assays reported may be skewed by the growth rates of the cells. Effects of certain mutations on senescence may be very similar, but due to slow growth, the curves look different, For example, the first occurrence on Fig. 1B, the mre11D yku80D vs the mre11D yku80D sir4D curves (a key result of the paper). The authors interpret this to say that loss of sir4 inhibits the senescence of the mre11D yku80D cells. To this reviewer, the data do not look like that, rather loss of sir4 suppresses the slow growth phenotype, with senescence occurring at about the same time and to the same extent. Compare the burgundy line with the dark yellow line: there is no difference in senescence, occurs for both at passage 3. Its just that the yellow cells grow better, and the curve is higher in the graph. This is a systematic problem for the liquid senescence assay that must be controlled adequately. Most people now use a equalized # of generations in the timeline, not absolute time as here. Therefore, throughout the paper, all the data from the liquid senescence assays must be re-evaluated!

  • Another problem for those graphs is a very poor growth phenotype initially that then gets dramatically better with time (for example the mre11D yku80D rad51 cells in Fig. 1D). These cultures do not show a senescence pattern at all, just continuously faster growth, which renders senescence assays virtually impossible. To this reviewer, the most likely interpretation for those curves is an accumulation of suppressor mutations, due to the very strong selective pressure.
    The data on Paf1 and Ctr9 are very clear but not really new.

  • The fact that a paf1 deletion causes a strong increase in direct repeat recombination is known since a long time (see Chang. M et al.; Mol. Cell. Biol. 19(2):1056-67; 1999). Therefore, this is confirmed here.

  • In terms of the description of the effects of Paf1, Ctr9 and Mps3, I request that the descriptions remain within the framework of what is measured and described in the work here. Particularly the Paf1/Ctr9 mutations have extremely pleiotropic effects with many, many processes being affected. Therefore, to ascribe the effect measured here to a downregulation of TERRA is an interpretative stretch or speculation and not supported by any data. Here for example, the authors would have to measure TERRA expression in all the mutant setups, plus for example an overexpression of TERRA should reverse the phenotype. The same is true for the loss of nuclear envelope localization; these are possibilities, but I could think of many others that are equally possible. Therefore, those speculations should not be placed in the results section, but rather labeled as such and put into the discussion.

  • Finally, in terms of the discussion of the survivor pathway, the authors should consider the newly described way how survivors arise in budding yeast (see Kockler et al.; Mol Cell Feb 19:S1097-2765(21)00090-3. doi: 10.1016/j.molcel.2021.02.004; 2021). The Type I vs Type II descriptions may have to be reconsidered and only one unified pathway applies. It would also be interesting to consider discussing how both Rif1 and Rif2 appear to limit telomerase access to telomeres but have opposing effects on recombination here. Further, for the model Fig 6, please include Tel1 as a mediator for telomerase recruitment. Without MRX, there also is no Tel1 kinase association.

Author Response

Response to Reviewer 1 Comments

The maintenance of telomeres ensures chromosome stability and as such is essential for continued cellular division. Given that this mechanism has importance for cancer cells in humans, it is of interest to investigate it in molecular detail. In this manuscript, telomere maintenance is probed in budding yeast cells using a variety of genetic deficiencies and measuring cellular senescence as well as subtelomeric DNA amplification. The project starts with the observations made many years ago that even in fully telomerase positive cells, a combined loss of the repair proteins yKu and Mre11 causes a phenotype that is indistinguishable from that of a loss of telomerase. It is assumed, but to date not directly shown, that this combination of mutations leads to a complete lack of recruitment of telomerase to telomeres, hence the phenotype, even though the enzyme itself still is present in cells.

 Here then, the influence of additional mutations on this mechanism is assessed. The results show increased subtelomeric recombination in cells that also harbour a deletion of Sir4 or Sir3, but not with a Sir2 deletion. Furthermore, while deleting Rif1 also renders telomeric recombination more efficient, a loss of Rif2 causes the opposite. All of these effects on recombination are dependent on the Rad51 and Rad52 genes, as one would expect. Furthermore, deleting Paf1 or Ctr9 dramatically increased the subtelomeric recombination. The authors also assessed the involvement of Siz2 or Mps3 in these effects, but only detected minor differences. Together, the data are discussed as showing a critical role for Sir3/Sir4 in maintaining a normal repressive subtelomeric environment. Quantitatively, the effect of absence of PAF-complex members causes an even more important disturbance/upregulation of recombination, but it is difficult to attribute this to a specific mechanism.

 Overall, the data generated are of good quality, appear robust and are well documented. Wherever possible, quantitative approaches were used, allowing for more telling conclusions. Below I outline some important suggestions for improvements at the conceptual level. However, please note that the obvious need to describe extensive and complicated genetic setups combined with a somewhat limited English makes for a difficult read. Here, I do not list all the shortcomings, it would take up too much space, but the manuscript definitely needs a major and thorough correction from a native English speaker.

Specifics: 

I do appreciate the liquid senescence assays; the best way to measure this parameter for yeasts. However, the assay does have some pitfalls as well. Here, certain of the assays reported may be skewed by the growth rates of the cells. Effects of certain mutations on senescence may be very similar, but due to slow growth, the curves look different, For example, the first occurrence on Fig. 1B, the mre11D yku80D vs the mre11D yku80D sir4D curves (a key result of the paper).

The authors interpret this to say that loss of sir4 inhibits the senescence of the mre11D yku80D cells. To this reviewer, the data do not look like that, rather loss of sir4 suppresses the slow growth phenotype, with senescence occurring at about the same time and to the same extent. Compare the burgundy line with the dark yellow line: there is no difference in senescence, occurs for both at passage 3. Its just that the yellow cells grow better, and the curve is higher in the graph. This is a systematic problem for the liquid senescence assay that must be controlled adequately. Most people now use a equalized # of generations in the timeline, not absolute time as here. Therefore, throughout the paper, all the data from the liquid senescence assays must be re-evaluated! 

Author response:

We thank reviewer for the expert comments. In the liquid senescence assay (Fig.1B), even at the first passage, most yku80Δmre11Δ cells stopped to divide due to permanent cell cycle arrest with very short telomeres (Supplementary Fig. S15A). This finding is further supported by the data that yku80Δmre11Δrad51Δ cells barely proliferate at the first passage in liquid medium, while growth seen in later successive passages is from the growth of survivors with highly amplified subtelomereic Y’ element. This explains why at the first passage the cell density of yku80Δmre11Δ cells from a single spore of tetrad dissection with initial conc. of A600=0.01 reaches to ~A600nm=3.6 after about 20 h growth at 30 Celsius in 220 rpm, and in contrast, WT spores with the same initial conc. reach about 7.0.

In addition, we also performed cell cycle analysis using spores of WT, yku80Δmre11Δ, and yku80Δmre11Δsir4Δ cells (kindly refer to the data figure below). In the flow cytometry analysis results, yku80Δmre11Δ or cells with different genotypes showed four categories (Figure A below), and >95% of cells analyzed (>1 million cells) (P7), therein, >55% are in G2/M phase (Fig B, lower panel) compared to ~20% in WT cells. These results suggest that yku80Δmre11Δ cells underwent G2/M phase arrest as senescence.

Consistently, yku80Δmre11Δsir4Δ showed ~38% cells at G2/M phase, suggesting that SIR4 deletion partially rescues senescence from G2/M phase arrest of yku80Δmre11Δ cells.

Figure legend: Cells in each subculture were stained with SYTOX Green Nucleic Acid Stain (Cat#: S7020, Invitrogen) for cell cycle analysis as described previously (Haase &Reed 2002 Cell Cycle). A calculated quantity of cells in overnight culture for each genotype was diluted into 10 ml fresh YPD medium to give an initial concentration of A600nm=0.01. Cell density (A600nm) was measured by a spectrophotometer, calculated and re-diluted into 10 ml fresh YPD medium approximate every 20 h. Approximately 1x107 cells were harvested by centrifugation for 2 min at 1000 × g, and then re-suspended in 1.5 ml of H2O followed by a slow addition and mixing of 3.5 ml 100% ethanol to fix the cells for overnight at 4°C. Following centrifugation for 2 min at 1000 × g, washing in 1 ml of H2O and transferring into a microcentrifuge tube, the fixed cells were resuspended in 0.5 ml RNAse solution (2 mg/ml RNAse A in 50 mM Tris pH 8.0, 15 mM NaCl) and incubated for 2 h at 37°C, before being collected and resuspended directly in 0.2 ml protease solution (5 mg/ml pepsin, 4.5 μl/ml concentrated HCl, in H2O). After incubation for 20 min in protease solution at 37°C, cells were centrifuged and re-suspended in 0.5 ml 50mM Tris-HCl pH 7.5 for storage at 4°C. For analysis, 50 μl of cell suspension was placed into 1 ml of SYTOX Green solution (1 μM SYTOX Green in 50 mM Tris pH 7.5) for 30 min and followed by sonication at low power, and standard flow cytometry analysis was done by BD LSRFortessa™ cell analyzer. The BL1 (530/30) channel was used to collect SYTOX fluorescence and data were analysed by the BD FACSDiva software.

Another problem for those graphs is a very poor growth phenotype initially that then gets dramatically better with time (for example the mre11D yku80D rad51 cells in Fig. 1D). These cultures do not show a senescence pattern at all, just continuously faster growth, which renders senescence assays virtually impossible. To this reviewer, the most likely interpretation for those curves is an accumulation of suppressor mutations, due to the very strong selective pressure. 
The data on Paf1 and Ctr9 are very clear but not really new. 
The fact that a paf1 deletion causes a strong increase in direct repeat recombination is known since a long time (see Chang. M et al.; Mol. Cell. Biol. 19(2):1056-67; 1999). Therefore, this is confirmed here. 

Author response:

In the liquid senescence assay, mre11∆ yku80∆ rad51 spore cells barely grow at the first passage (Fig. 1D), suggesting that most of yku80Δmre11Δ cells are experiencing senescence, with some cells escaped senescence via sub-telomeric recombination. It is also possible that following successive passages, these senescence-escape cells maintained their telomeres successfully and gradually took up the proliferate advantage.

In terms of the description of the effects of Paf1, Ctr9 and Mps3, I request that the descriptions remain within the framework of what is measured and described in the work here. Particularly the Paf1/Ctr9 mutations have extremely pleiotropic effects with many, many processes being affected. Therefore, to ascribe the effect measured here to a downregulation of TERRA is an interpretative stretch or speculation and not supported by any data. Here for example, the authors would have to measure TERRA expression in all the mutant setups, plus for example an overexpression of TERRA should reverse the phenotype. The same is true for the loss of nuclear envelope localization; these are possibilities, but I could think of many others that are equally possible. Therefore, those speculations should not be placed in the results section, but rather labeled as such and put into the discussion. 

Author response:

We thank reviewer for the comments. In the results section, in line329, we have accordingly rephrased the wording. Since TERRA regulated by Paf1/Ctr9 or Sir4, or telomere perinuclear localization regulated by Sir4, Siz2, Mps3 and Sir4-Mps3 interaction, have been well documented in literature, we provide relevant information in the Introduction section, with speculations discussed in the Discussion section.

Finally, in terms of the discussion of the survivor pathway, the authors should consider the newly described way how survivors arise in budding yeast (see Kockler et al.; Mol Cell Feb 19:S1097-2765(21)00090-3. doi: 10.1016/j.molcel.2021.02.004; 2021). The Type I vs Type II descriptions may have to be reconsidered and only one unified pathway applies. It would also be interesting to consider discussing how both Rif1 and Rif2 appear to limit telomerase access to telomeres but have opposing effects on recombination here. Further, for the model Fig 6, please include Tel1 as a mediator for telomerase recruitment. Without MRX, there also is no Tel1 kinase association.

Author response:

We appreciate reviewer comments. We have accordingly added Tel1 as a mediator for telomerase recruitment in Fig. 6B.

Reviewer 2 Report

In this manuscript the authors studied the role of SIR proteins and RIF1/2 in subtelomere recombination and the reversal of senescence in yeast in the presence of telomerase. They showed that the senescence-like phenotype induced by the double deletion yKU80 and MRE11 is rescued by SIR3 and SIR4 deletions but not by SIR2 deletions. The results are clearly presented and supported by experimental data.

Minor points:

  1. In figure 3, the length of the TRF signal should be included, either at the bottom of the gel or as a small table.

  1. It is not clear why the title ‘Rad51 is required for senescence evasion induced by Sir4 deficiency’ does not include RAD52, which is the first experiment the authors showed.

  1. The paragraph starting in line 152 is difficult to read, that needs to be either split or rephrase.

  1. The explanation of the yku80-135i allele in line 245 is needed earlier in the text.

  1. English needs to be corrected in line 54.

Author Response

Response to Reviewer 2 Comments

In this manuscript the authors studied the role of SIR proteins and RIF1/2 in subtelomere recombination and the reversal of senescence in yeast in the presence of telomerase. They showed that the senescence-like phenotype induced by the double deletion yKU80 and MRE11 is rescued by SIR3 and SIR4 deletions but not by SIR2 deletions. The results are clearly presented and supported by experimental data.

Minor points 

  1. In figure 3, the length of the TRF signal should be included, either at the bottom of the gel or as a small table.

Author response:

We thank reviewer’s comments, and have accordingly marked the median length of XhoI-TRF fragments for two replicates of each strain with a white dash line.

  1. It is not clear why the title ‘Rad51 is required for senescence evasion induced by Sir4 deficiency’ does not include RAD52, which is the first experiment the authors showed.

Author response:

We thank reviewer. Rad51 is required for subtelomeric Y’ element recombination specifically, whereas Rad52 is a general mediator of homologous recombination. We agree that further studies are required.

  1. The paragraph starting in line 152 is difficult to read, that needs to be either split or rephrase.

Author response:

We thank reviewer’s comments and have accordingly revised this.

  1. The explanation of the yku80-135i allele in line 245 is needed earlier in the text.

Author response:

We appreciate reviewer’s suggestion, and have accordingly moved the explanation of the yku80-135i allele to the first paragraph of the Introduction section, line38.

  1. English needs to be corrected in line 54.

Author response:

We have accordingly revised this part in line54-55.